# Observational Evidence of the Vertical Exchange of Ozone within the Urban Planetary Boundary Layer in Shanghai, China

Yixuan Gu [1,2], Fengxia Yan [3], Jianming Xu [1,2,*], Liang Pan [1,2], Changqin Yin [1,2], Wei Gao [1,2] and Hong Liao [4]

[1] Shanghai Typhoon Institute, Shanghai Meteorological Service, Shanghai 200030, China; yixuan.gu@colorado.edu (Y.G.); panlg@163.com (L.P.); yincq@gd121.cn (C.Y.)

[2] Shanghai Key Laboratory of Meteorology and Health, Shanghai Meteorological Service, Shanghai 200030, China

[3] East China Air Traffic Management Bureau, Shanghai 200135, China

[4] Jiangsu Collaborative Innovation Center of Atmospheric Environment and Equipment Technology, Jiangsu Key Laboratory of Atmospheric Environment Monitoring and Pollution Control, School of Environmental Science and Engineering, Nanjing University of Information Science and Technology, Nanjing 210044, China; hongliao@nuist.edu.cn

[*] Correspondence: metxujm@163.com

**Abstract:** The vertical mass exchange of ozone ($O_3$) plays an important role in determining surface $O_3$ air quality, the understanding of which, however, is greatly limited by the lack of continuous measurements in the vertical direction. Here, we characterize $O_3$ variations at a high-altitude monitoring site at the top of Shanghai Tower (SHT) and discuss the potential impacts of the vertical exchange of air pollutants on $O_3$ air quality within the urban planetary boundary layer (PBL) based on continuous measurements during 2017–2018. During the daytime, two distinct patterns of vertical $O_3$ gradient are detected. In summer, the daytime $O_3$ formation at SHT is observed to be more limited by nitrogen oxides ($NO_x$) than the surface, which, together with the efficient vertical mixings, results in higher $O_3$ levels in the upper mixing layer. In cold months, the opposite vertical gradient is observed, which is associated with weak vertical exchange and $NO_x$-saturated $O_3$ formation. A nighttime $O_3$ reservoir layer and consistent morning $O_3$ entrainments are detected all year round. These results provide direct evidence of the vertical mixings within the urban PBL, underscoring the pressing need for improving vertical resolution in near-surface layers of air quality models.

**Keywords:** ozone; vertical exchange; planetary boundary layer; tower observation; Shanghai

## 1. Introduction

Ozone ($O_3$) pollution has become an urgent environmental problem in China, posing a serious threat to air quality, human health, land ecosystem, and climate change [1–4]. In polluted regions, surface $O_3$ is generated by the photochemical oxidation of volatile organic compounds (VOCs) and carbon monoxide (CO) in the presence of nitrogen oxides ($NO_x \equiv NO + NO_2$). The production process is highly nonlinear, and ambient $O_3$ concentrations can be sensitive to changes in precursor emissions in terms of both magnitude and sign. To alleviate air pollution, stringent emission control measures have been implemented by the Chinese government through the Clean Air Action [5]. However, the sharply reduced emissions of $NO_x$ led to an unexpected increase in surface $O_3$ levels since $O_3$ formation is generally in the $NO_x$-saturated regime [6,7]. During 2013–2017, the observed daily maximum 8 h averaged (MDA8) $O_3$ concentrations increased at a rate of 1–3 ppb yr$^{-1}$ in eastern China [8]. Most severe $O_3$ pollution occurred in urban agglomerations, which are densely populated, resulting in elevated exposure of the population and increasing damages to public health [9]. Therefore, understanding the processes that influence $O_3$ accumulation in cities has become one of the top concerns in dealing with air pollution problems in China.

Tropospheric $O_3$ has a lifetime ranging from a few hours to weeks in polluted urban areas, which can thus be transported at difference time and spatial scales. The synthetic effect of nonlinear $O_3$ formation and complex horizontal and vertical transport of $O_3$ and its precursors within the planetary boundary layer (PBL) significantly increase the difficulties of controlling $O_3$ pollution in cities. In addition to serving as hotpots for anthropogenic emissions of $NO_x$ and VOCs, cities also have unique canopy structures which modify the thermal and dynamical characteristics of air, leading to changes in meteorological conditions, which then influence the physical (e.g., advection, vertical mixing, and deposition) and chemical (e.g., photochemical reaction) behaviors of air pollutants [10–13]. For instance, elevated temperature and increased roughness near the urban surface were proved to greatly modify local circulations in cities, which accelerate chemical conversions, suppress horizontal lower boundary layer transport, and enhance vertical mixings within the PBL [10,14–17]. A large number of studies have investigated the impact of city-modified circulations on regional air quality, suggesting that enhanced vertical mixings tend to decrease the concentrations of primary air pollutants (e.g., $NO_x$, CO, and primary organic aerosols) while increasing the concentrations of secondary air pollutants (e.g., $O_3$ and secondary organic aerosols) near the surface [11,16,18–22].

As a result of suppressed horizontal transport and enhanced vertical mixing, concentrations of air pollutants usually exhibit a sensitive response to the diurnal variations in PBL height in cities. The case of $O_3$ is even more complicated than that of other species (e.g., $PM_{2.5}$), since the boundary layer dynamics affect not only $O_3$ but also its precursor, thus resulting in more nonlinear changes in $O_3$ concentrations. Vertical $O_3$ exchanges are usually considered to aggravate surface air pollution [10,12,23–25]. $O_3$ from the daytime mixing layer (ML) can also be maintained at the nocturnal residual layer (RL) in the absence of advection when the PBL decreases after sunset. Due to a lack of emission sources, aloft $O_3$ cannot be completely titrated and consumed by nitric oxide (NO) and other reductive pollutants at night, resulting in a reservoir of $O_3$ at a few hundred meters above the surface. The high level of $O_3$ in the nighttime RL exerts a significant influence on nighttime heterogeneous chemistry (e.g., secondary aerosol formation) [26–28]. $O_3$-rich air may also mix down to the surface, triggering a nocturnal $O_3$ enhancement under favorable weather conditions [29,30]. As the PBL develops after sunrise, the preserved $O_3$ from the RL is regarded as an important contributor of morning $O_3$ accumulation near the surface, since the increasing surface temperature and the formation of the ML accelerate the vertical exchange of $O_3$ between the RL and the surface, especially under clear skies and weak surface wind conditions [13,19,24,31–36]. By analyzing measurements obtained from 220 ozonesondes in Houston, Morris et al. [32] indicated that the morning RL $O_3$ concentrations explained 60–70% of the variability observed in the afternoon ML $O_3$ during July 2004–June 2008. Zhu et al. [37] suggested that more abundant $O_3$ at the RL would lead to higher concentrations of $O_3$ near the surface. Based on two cases of sounding measurements in the summer of 2012 and 2013, their calculations suggested that the downward transport of the RL $O_3$ contributed to more than 50% of the surface $O_3$ during 06:00–10:00 LST (local standard time) in Beijing compared to those produced by photochemical reactions.

Though the impacts of enhanced vertical mixings on urban $O_3$ air quality have been reported in many cities, direct observations of the upper PBL $O_3$, especially those during nighttime, are extremely limited. Previous observational studies mostly focused on cases [27,32,37,38], where observations are either typically scheduled for the afternoon (e.g., ozonesond), operated at a low frequency (e.g., aircraft and balloon), or limited by the retrieval algorithm and instrument coverage (e.g., satellite and lidar). Tower-based measurements are thus considered the most effective approaches to obtain continuous $O_3$ observations in the upper PBL with a high accuracy. Compared to densely conducted surface measurements, tower-based $O_3$ monitoring sites are quite limited in China. Based on continuous tower measurements, vertical distributions, the seasonal/decadal changes in $O_3$, and the formation of surface nighttime $O_3$ enhancement were examined in Beijing [39–41], Tianjin [39,42], and Guangzhou [43–45], respectively, with the highest measurements con-

ducted at 488 m. In this study, we present the first analysis of continuous upper-layer $O_3$ measurements conducted at a high-altitude opening observatory at the top of Shanghai Tower with a monitoring height of 600 m. The vertical $O_3$ exchange in/between different PBL structures in Shanghai is investigated by comparing the 2-year $O_3$ and $NO_x$ measurements at the Shanghai Tower site (referred to as SHT in the following) to those obtained simultaneously at a surface monitoring site at Pudong (PD). The impacts of different vertical exchange patterns on surface $O_3$ air quality and related mechanisms are discussed, aiming to provide additional observational evidence and implications for surface $O_3$ pollution control in cities.

## 2. Materials and Methods

### 2.1. The Surface and Tower-Based Measurements

We conducted continuous observations of $O_3$, NO, and nitrogen dioxide ($NO_2$) at an opening observatory at the top of Shanghai Tower (121.512° E, 31.239° N, 632 m) from 1 January 2017 to 31 December 2018. Shanghai Tower is Asia's tallest and the third tallest building in the World by height to architectural top, located at Lujiazui, the financial center of Shanghai. Compared to heights where other tower observations (e.g., 280 m in Beijing [39–41]; 220 m in Tianjin [39,42]; 450 m in Guangzhou [43–45]; 160 m in London [46]) are conducted, SHT (~600 m) is higher, at about 1/2 of the daytime ML (~800–1200 m) [47,48]. The relative location of the site to the top of the boundary layer varies during different stages of PBL evolution (Section 3.1). The observations thus provide unique observational evidence of the $O_3$ characteristics in the ML, RL, and nocturnal boundary layer (NBL) during different times of day. To investigate the vertical exchanges of $O_3$ and its precursors between the surface and RL, and in the convective mixing layer, $O_3$ and $NO_x$ measurements were simultaneously collected at a surface monitoring site (121.559° E, 31.226° N, ~9 m) at PD, Shanghai. The PD site (referred to as PD in the following) is located ~4.7 km southeast from SHT in the same district, where the observed air pollutant concentrations were used to represent the surface pollutant levels over SHT and surrounding areas [49].

At both PD and SHT, hourly $O_3$ concentrations were measured using the Model 49*i* Ozone Analyzer (Thermo Fisher Scientific Inc., Waltham, MA, USA) based on the absorbance of ultraviolet light by ozone molecules at a wavelength of 254 nm. Hourly $NO_x$ concentrations were measured by a chemiluminescent trace level analyzer (Model 42*i*-TL, Thermo Fisher Scientific Inc., Waltham, MA, USA) with a detection limit of 0.025 ppb. All instruments met the technical specifications for the United States Environmental Protection Agency, with a quality control check performed every 3 d. The filter was replaced every 2 weeks and calibrated every month. During the observational period, the capture rates of the qualified $O_3$ and $NO_x$ measurements were 90.9% and 92.4%, respectively. To investigate the influence of wet deposition on our results, the observational results are compared with those considering only nonrainy periods. The qualified records were selected according to the simultaneous surface measurements of meteorological parameters, including temperature, relative humidity, horizontal wind speed, wind direction, and precipitation at PD. The geographical locations of the monitoring sites are displayed in Figure 1.

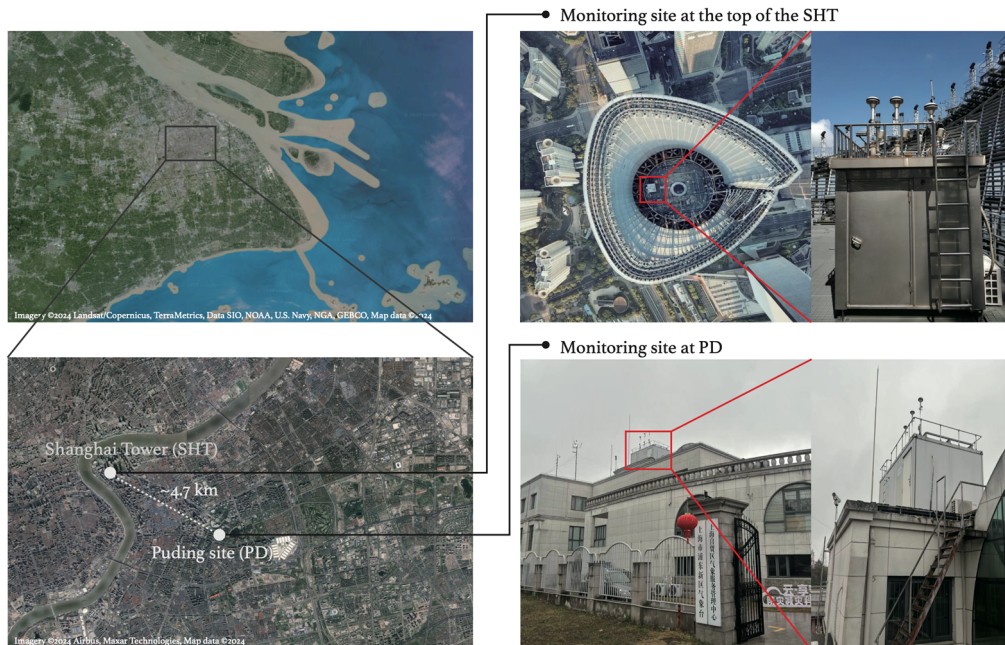

**Figure 1.** The geographic location, landscape, and deployment of the monitoring sites at Shanghai Tower (SHT, high-altitude) and Pudong (PD, surface), respectively.

### 2.2. The PBL Height Reanalysis Data

To analyze the seasonal variations in the PBL height, hourly PBL height data during the observational period (i.e., 2017–2018) were obtained from the fifth-generation reanalysis datasets developed by the European Center for Medium-Range Weather Forecasts (ERA5, https://cds.climate.copernicus.eu/cdsapp#!/dataset/reanalysis-era5-single-levels, last access: 20 December 2023), with a horizontal resolution of 0.25° × 0.25° [50]. Being compared with many radiosonde-based observations, the ERA5 reanalysis PBL height has been reported to be able to capture the diurnal and seasonal cycle of the PBL structure [51–53]. For deep boundary layers (>1000 m), the uncertainties of the reanalysis data are generally within 20%. Based on five-year (2010–2015) lidar measurements deployed at PD, Pan et al. [47] examined the diurnal and seasonal variations in PBL height exactly over the studied urban region, suggesting that the daytime PBL height ranged from ~800 m in winter to ~1200 m in summer, while the NBL heights were generally shallower than 200 m. We thus assume that the daytime ML can be well reflected by the ERA5 data. The urban NBL heights, however, were usually observed to range from approximately 100 to 400 m [47,54], in which case the uncertainties of the reanalysis data can exceed 50%. The reanalysis PBL heights were still reported to be able to capture seasonal variations when multi-year averages were considered [51]. Therefore, we only considered the two-year mean values to identify the PBL height in our analysis. By making comparisons with remote sensing retrievals, recent studies have indicated that the ERA5 which estimated the NBL heights did not reflect the influence of the RL [55,56]. We infer the location of the RL by comparing the height of the NBL to that of the daytime ML, as the top of the ML can generally become the top of the leftover when the PBL transited from unstable in the daytime to stable at night [53].

## 3. Results and Discussion

### 3.1. The Location of the SHT Site

The earlier observational evidence indicates that the layer (~600 m) where the SHT site is located (referred to as the SHT layer in the following) is in the upper ML during the daytime, while it is above the top of the PBL during the nighttime. When the RL is formed, the SHT layer is more likely to be included in the RL. The observations are thus able to represent air pollutant levels in various PBL structures during different times of day. The

reanalysis data provide additional evidence of the relative location of SHT to the ML, NBL, and RL. Figure 2 displays the diurnal variations in the reanalysis PBL height in the pixel where SHT is located. The reanalysis PBL heights present comparable magnitudes and similar diurnal patterns as those derived from sounding data [48,57]. In all the seasons, the PBL heights start to increase at 07:00–08:00 LST, achieving peak values (~800–1200 m) at 13:00–14:00 LST and decreasing to 200–300 m after sunset. The deepest daytime PBL appears in October since the prevailing synoptic of the continental high pressure favors the development of the convective ML. SHT is included in the urban PBL from 10:00 to 16:00 LST as a result of vigorous turbulent mixing, when the SHT measurements can be used to analyze $O_3$ characteristics at the upper ML. During other times of day, SHT is observed to be mostly above the top of the reanalysis PBL. The SHT observations, especially those taken during the nighttime, tend to reflect air pollutant levels in the RL or even the free atmosphere. By comparing the observed $O_3$ variations at the surface (PD) and SHT layers, we are thus able to obtain direct observational evidence to investigate vertical $O_3$ exchanges, for example, the $O_3$ entrainments from the RL to the surface in the morning and the strong vertical $O_3$ mixings during noontime, over this urban area.

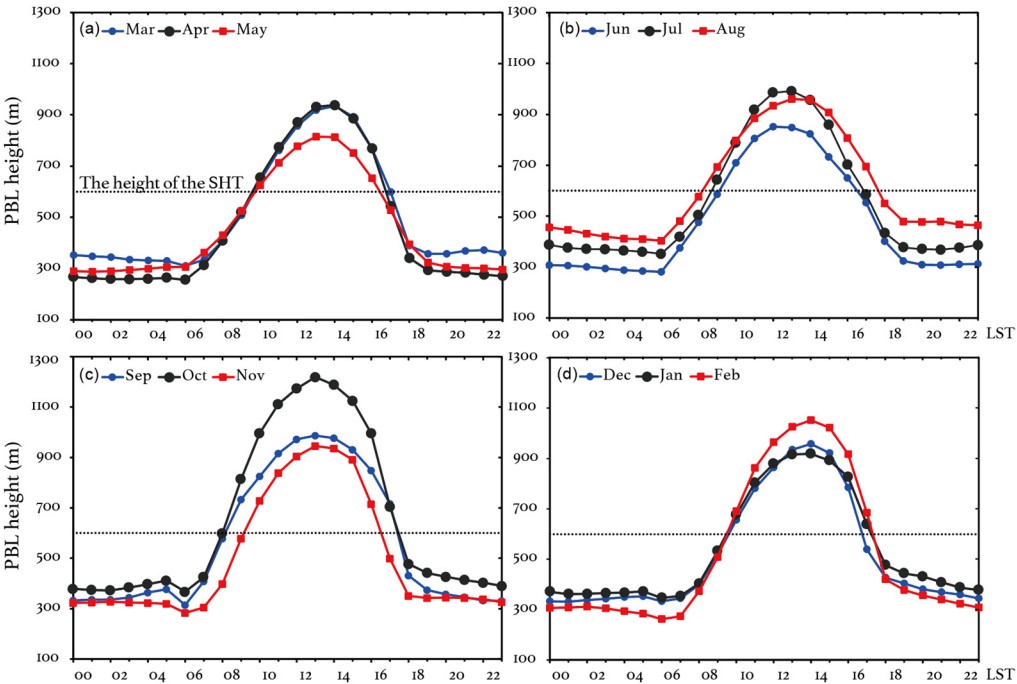

**Figure 2.** Diurnal variations in the reanalysis PBL height (unit: m) at the grid box where the Shanghai Tower site (SHT) is. The black dash lines represent the altitude (~600 m) of SHT. The results for (**a**) spring (March to May), (**b**) summer (June to August), (**c**) autumn (September to November), and (**d**) winter (December to February) are shown. Numbers in the *x*-axis are local standard time (LST).

### 3.2. Observed $O_3$ Characteristics at the Surface and SHT

#### 3.2.1. Monthly Variations in $O_3$ at SHT and the Surface

Figure 3a shows the observed monthly variations in $O_3$ concentrations at SHT and PD. Generally, SHT $O_3$ exhibits similar seasonal patterns to those at PD, presenting higher concentrations in warm seasons but lower concentrations in winter. The observed seasonal variations are consistent with the general variations in solar radiation throughout the year, indicating that $O_3$ production is dominated by the photochemical processes at both the surface and upper PBL. Compared to the surface measurements, SHT $O_3$ exhibited higher concentrations, with an observed mean level of 40.0 ppbv during the observational period, 23.8% higher than that (32.3 ppbv) at PD. As the relative location of SHT to the PBL changes (Section 3.1), the relationship of $O_3$ between SHT and the surface is different when the layer is included in various PBL structures. Figure 3b,c further display the monthly $O_3$

variations at PD and SHT, averaged over daytime and nighttime, respectively. The daytime observations are selected from 09:00 to 16:00 LST, when SHT is generally included in the convective ML (Figure 1). Compared to the nighttime (22:00–05:00 LST) observations, the daytime $O_3$ concentrations at SHT show better consistency with those at PD. The mean $O_3$ concentrations are 42.3 and 42.0 ppbv at SHT and PD, respectively. Consistency can be achieved since air pollutants tend to be well mixed in a fully developed convective ML. It is noted that the observed SHT $O_3$ exhibits higher concentrations in summer and early autumn compared to the surface ones, but it presents negative deviations in spring and winter. The results indicate a different mechanism of $O_3$ accumulation at the surface and upper ML in various seasons. During nighttime, the observed $O_3$ exhibits more pronounced discrepancies between the two sites (Figure 3c), when the observed mean $O_3$ concentration (38.4 ppbv) at SHT is 51.8% higher than that (25.3 ppbv) at PD. As SHT is generally above the top of the shallow NBL (Figure 1), air pollutants at SHT are more likely to be isolated from those near the surface at night. The large nighttime discrepancies thus can be explained since SHT $O_3$ tends to be less titrated and consumed. Once the RL forms under favorable weather conditions (e.g., weak winds and strong inversion), the $O_3$-rich air would be able to be stored and entrain to the surface again as the convective ML develops in the subsequent morning. To investigate the impact of precipitation on the $O_3$ characteristics, we compare the seasonal variations with those considering measurements only taken during nonrainy days. The results (Figure S1) suggest that the seasonal patterns are barely affected by rain.

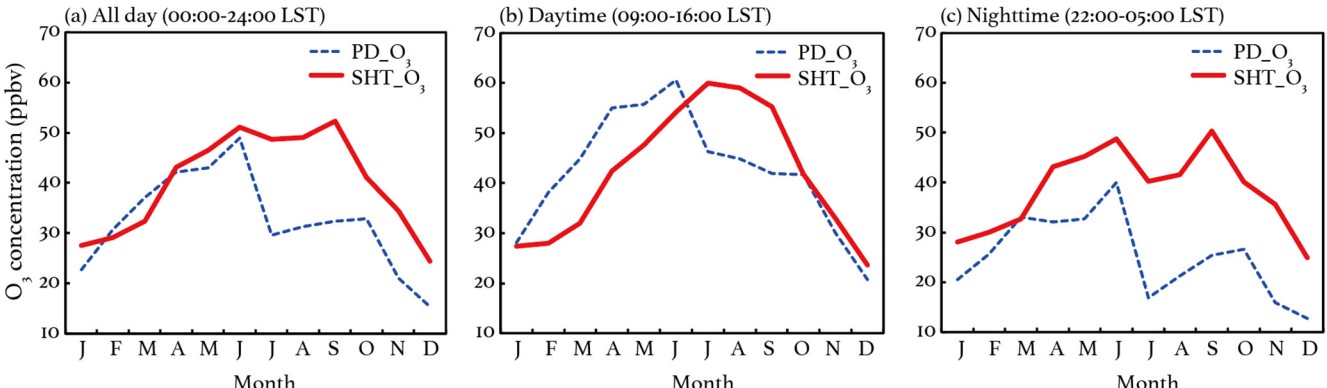

**Figure 3.** The observed monthly mean $O_3$ concentrations (unit: ppbv) at SHT and PD during 2017–2018. The results for (**a**) all-day (00:00–24:00 LST), (**b**) daytime (09:00–16:00 LST), and (**c**) nighttime (22:00–05:00 LST) averages are presented.

3.2.2. Diurnal Variations in the $O_3$ Differences between SHT and the Surface

To investigate the potential mechanism resulting in the observed $O_3$ differences at the surface and upper PBL, we calculate the $O_3$ difference (OD), defined as the observed seasonal mean $O_3$ concentrations at SHT minus those at PD. The mean diurnal variations in $O_3$ in each season are shown in Figure 4. SHT $O_3$ exhibits a similar diurnal pattern to that in PD in summer (June to August, JJA), while the diurnal variations it presents in other seasons are not distinctive, especially in winter (December to January, DJF). The results suggest that $O_3$ at SHT can be more affected by surface pollutants in summer than in other seasons, indicating more significant vertical mixings during these two layers. The calculated OD exhibits positive peaks in the morning and evening, while it presents a valley or a negative peak during the noontime, which aligns well with the diurnal variations in the urban PBL. The observed OD starts to decrease at 05:00–06:00 LST when the convective ML begins to develop and increase again in the afternoon as the turbulence becomes weaker. These changes suggest a period when the vertical exchanges between the surface and SHT layer occur. As the ML fully develops near the noontime, $O_3$ tends to exhibit the highest homogeneity in the vertical direction within the urban PBL, which is reflected by

the relatively small differences in $O_3$ levels between the surface and SHT during this period. Once the inversion is established after sunset, the high-level $O_3$ produced or dispersed in the daytime ML is likely to be stored at the SHT layer, since this layer is generally above the top of the NBL and can be a part of the RL, where air pollutants are isolated from the surface. In the absence of $NO_x$, $O_3$ at the SHT layer is less likely to be titrated and consumed at night compared to those near the surface, resulting in an increased OD between PD and SHT. Consistent with the shallow and stable NBL, the observed large OD sustains from 21:00 to 05:00 LST, ranging from 15 to 20 ppbv in summer, 19 to 23 ppbv in fall (September to November, SON), and 5 to 9 ppbv in winter and spring (March to May, MAM).

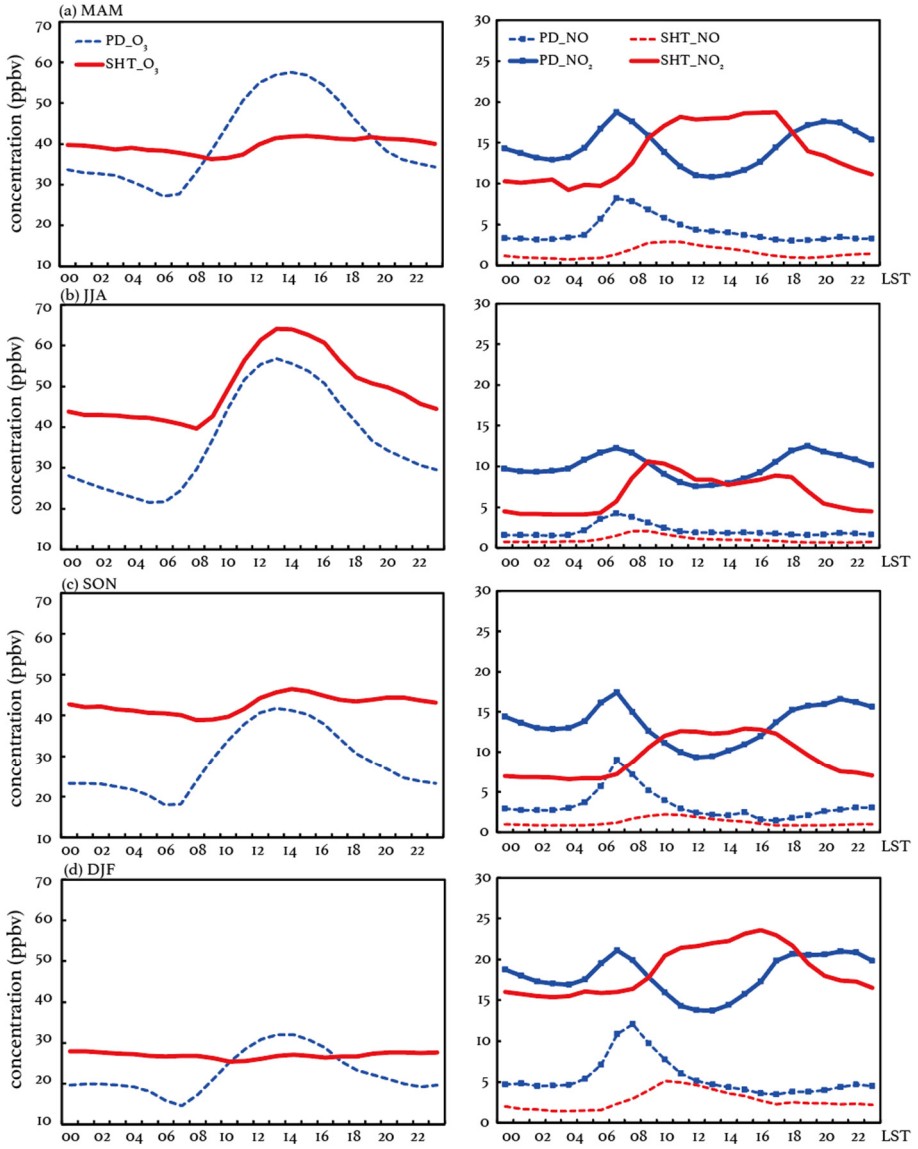

**Figure 4.** The seasonal mean diurnal variations in concentrations (unit: ppbv) of $O_3$, NO, and $NO_2$ at SHT and PD in (**a**) spring (MAM), (**b**) summer (JJA), (**c**) autumn (SON), and (**d**) winter (DJF) during 2017–2018.

Though the OD exhibits similar decreases from around 06:00 LST all year round, the $O_3$ concentrations at SHT are observed to be smaller than those at PD, resulting in negative OD values during 10:00–16:00 LST in DJF and MAM. The negative values suggest that the vertical exchanges during this period are more likely to result in upward $O_3$ flux, which helps to alleviate surface $O_3$ pollution. The $O_3$-rich air aloft is then able to be stored in the SHT layer, which is reflected by the observed peak (11.1–12.2 ppbv) of the positive $O_3$

deviations in the early morning. Vertical mixings after the destruction of the inversion layer thus contribute to the accumulation of $O_3$ near the surface. In JJA and SON, the OD is observed to be positive during both the daytime and nighttime, indicating that the vertical $O_3$ exchanges, if they occur, are more likely to aggravate surface $O_3$ pollution. The positive ODs present maximum values of 20.6–22.5 ppbv during 4:00–6:00 and minimum values of 3.5–4.6 ppbv during 09:00–12:00 LST. Compared to those in MAM and DJF, the observed OD in JJA and SON is undoubtedly more favorable for the surface $O_3$ enhancements during vertical mixing. As Figure 3 displays, the observed $O_3$ concentrations at SHT are generally consistent during the polluted season. The large OD can mainly be attributable to low surface $O_3$ concentrations (PD), indicating more vigorous $O_3$ depression, especially at night. The large nighttime differences may result in more $O_3$ entrainments from the RL to the surface during the morning, exerting more significant adverse impacts on surface $O_3$ air quality.

### 3.2.3. Diurnal Variations in the $O_3$ Changing Rate

To further investigate the causes of ODs in various seasons, we analyze the diurnal variation in the $O_3$ changing rate (OCR = d[$O_3$]/dt) at SHT and PD in Figure 5. The OCR can be determined by the changes in air pollutant concentrations due to transport, chemical production, chemical loss, and emissions. Since $O_3$ is a secondary pollutant which is not directly emitted into the atmosphere, d[$O_3$] induced by emission changes can be zero. The effect of advection can also be mitigated by the application of a monthly average for the diurnal trend analysis. The observed OCR at PD exhibits similar diurnal patterns as those reported in previous works [58]. Generally, a positive OCR is evident from 06:00 to 14:00 LST during summertime and 08:00 to 13:00 LST during wintertime, suggesting the accumulation of $O_3$ near the surface. Between 08:00 and 10:00 LST, the OCR reaches relatively high values due to enhanced $O_3$ production promoted by increasing precursor emissions and sunlight. Negative OCRs are observed after 15:00 LST, and these extend to the morning. This period is characterized by weakened photochemistry as the sun goes down, leading to an increase in the titration and consumption of $O_3$. Consequently, there is a prevalence of $O_3$ depressions compared to production near the surface. The most rapid $O_3$ decrease, ranging from −3.0 to −6.0 ppbv h$^{-1}$, at PD occurs during 17:00–19:00 LST, which is associated with a resurgence in $NO_x$ emissions during evening rush hours.

In contrast, the observed OCR at SHT exhibits more diversities in diurnal and seasonal variations, which has been seldom discussed in the literature. During October to May, the daytime OCR at SHT is consistently lower than that at PD. This distinction is particularly pronounced in winter, when the mean OCR ranges between 1.2 and 2.0 ppbv h$^{-1}$ at PD, contrasting with nearly zero values at SHT. In addition to lower magnitudes, the positive OCR at SHT stands out for its comparatively briefer duration during the daytime. Lasting 2 h in winter and 4–5 h in summer, the duration of the positive OCR at SHT is approximately one-third to one-half of the time observed at PD. The results suggest that the daytime $O_3$ production at the upper ML can be smaller than that near the surface, which explains the observed negative OD in DJF and MAM in Figure 4. During June to September, there is a notable increase in the daytime OCR at SHT. The largest enhancement is observed in July and August, when the positive OCR shows a duration of approximately 7 h, with peak values ranging from 8.7 to 9.7 ppbv h$^{-1}$. These values are comparable and even higher than those recorded at PD, where the peak OCR values range from 7.8 to 9.7 ppbv h$^{-1}$. The increased OCR indicates more favorable conditions for $O_3$ accumulation at SHT, which results in the observed positive OD in these months. The observed maximum daytime $O_3$ concentrations at SHT reach 68.3 ppbv in July, 15.1 ppbv higher than those at PD. The pronounced vertical $O_3$ gradient is highly likely to exert detrimental effects on the surface $O_3$ air quality, especially during periods of vertical $O_3$ mixing facilitated by favorable weather conditions (e.g., weak horizontal winds). The positive $O_3$ differences between the surface and upper PBL during the nighttime are also more likely to be preserved.

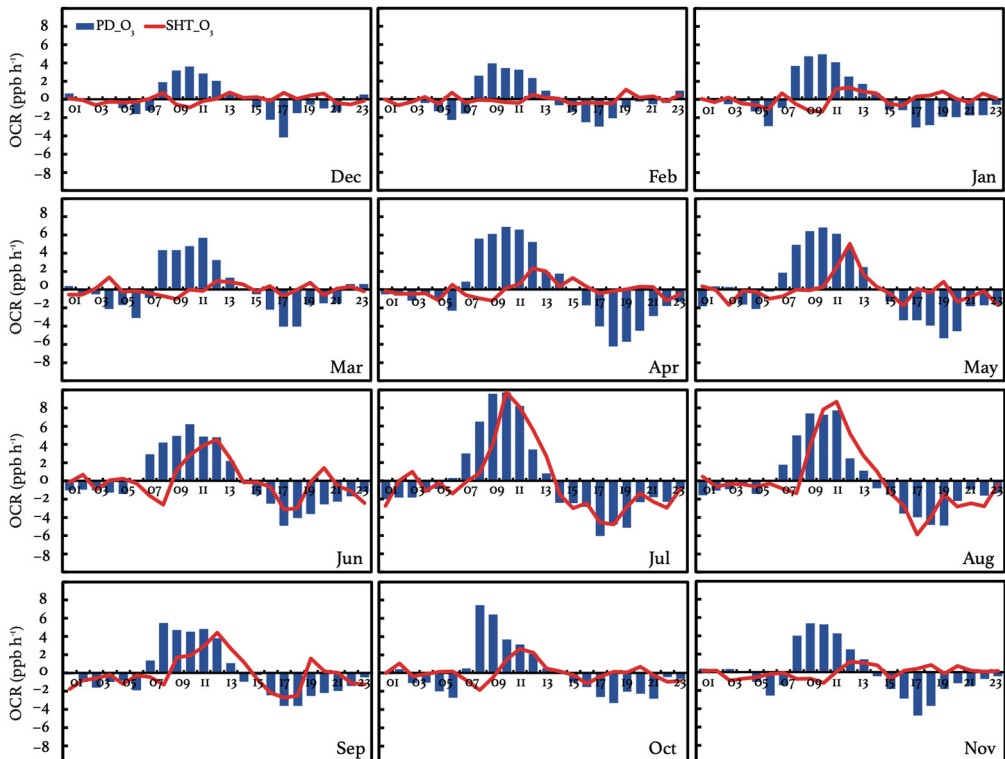

**Figure 5.** Monthly mean diurnal variations in $O_3$ changing rate (OCR = d[$O_3$]/dt, unit: ppbv h$^{-1}$) at SHT and PD from April to September averaged over 2017–2018.

At night, the OCR variance at SHT is generally smaller than that at PD, suggesting that less $O_3$ is depressed at the upper layer. Notably, a consistent negative OCR is detected during 05:00–07:00 LST at SHT, ranging from $-0.1$ to $-1.6$ ppbv h$^{-1}$. Concurrently, the observed OCR at PD begins to rise, exhibiting positive values ranging from 0.30 to 3.0 ppbv h$^{-1}$. In the early morning, $NO_2$ undergoes minimal photo-disassociation due to weak sunlight, resulting in limited $O_3$ production. The observed $O_3$ increases at PD and $O_3$ decreases at SHT thus can be largely attributed to vertical mass exchanges since the convective ML starts to develop. The vertical $O_3$ gradient leads to opposite changes in $O_3$ concentrations at SHT and PD. Since the surface $O_3$ is continuously depressed by the NO exhaust in the morning [58,59], the adverse impacts of the downward entrainments on the surface $O_3$ air quality can be partly offset, which is reflected by the less significant reflections in surface $O_3$ concentrations than those at SHT, as shown in Figure 5.

### 3.3. The Observed $O_3$-$NO_x$ Relationship

The vertical gradients of $O_3$ production play a critical role in the vertical $O_3$ exchanges, determining whether the $O_3$ flux is upward or downward within the urban PBL. Given the limited direct emissions of $O_3$ precursors at the upper layer, the monthly mean $O_3$ production at SHT is predominantly affected by the boundary layer dynamics and solar radiation. The former brings surface precursors to the upper layer, while the latter influences photochemistry. As SHT is close to the skyline of Shanghai, we assume that the daylight conditions there can be more beneficial to $NO_2$ photolysis than those near the surface due to less shielding of the urban canopy. However, the observed daytime OCR at SHT exhibits seasonally varying differences compared to that at PD. During cold months, daytime $O_3$ production tends to be inhibited at SHT, indicating faster $O_3$ accumulation near the surface than the upper ML, while in summer and early autumn, daytime $O_3$ production is promoted at SHT. The different $O_3$ productions result in a different OD, which brings opposite outcomes during the vertical daytime mixings.

To investigate the potential causes affecting daytime $O_3$ production at SHT, the seasonal mean diurnal variations in NO and $NO_2$ are examined in Figure 4 as well. The observed NO at SHT exhibits similar patterns of diurnal variations as those near the surface, yet peak values appear later. The surface NO concentrations reach their peak at around 7:00, aligning with the increasing NO emissions during the morning rush hour. In contrast, at SHT, the NO peaks appear at 08:00–09:00 LST in JJA and 10:00–11:00 LST in other seasons, exhibiting a lag of 1–4 h compared to the those near the surface. The observed $NO_2$ at SHT peaks simultaneously with the NO, which occurs 1–4 h later than those near the surface. For lack of direct emissions, the delayed NO and $NO_2$ peak at SHT can be mainly associated with the boundary layer dynamics. The surface $NO_2$ presents two peaks during the daytime, which is a typical diurnal pattern in urban areas. The first peak is driven by the increased $NO_x$ emissions in the morning, which aligns closely with the increases in NO. As a result of enhanced photolysis, the $NO_2$ concentrations at PD decrease quickly after the first peak, accompanied by increases in surface $O_3$ concentrations (Figure 4). After reaching the minimum during 12:00–14:00 LST, the surface $NO_2$ concentration re-increases due to weakened photolysis and reaches the second peak during the evening rush hour. In summer, $NO_2$ at SHT is observed to present a similar magnitude and variations as those near the surface after the appearance of the delayed morning peak. The observational evidence suggests a likely uniform mixing of air pollutants between the surface and the SHT layer. The more efficient upward transport of surface air pollutants is also supported by the significant enhancement of the OCR (Figure 5) and the earlier inclusion of SHT in the daytime convective ML (Figure 2). In MAM and DJF, the $NO_2$ concentrations are observed to be consistent after the morning peak, indicating relatively weak $NO_2$ photolysis and less effects of surface sources.

Even though the vertical exchange has been proved to be sufficient in summer, it can still not explain well the observed potential differences in daytime $O_3$ production between the surface and the SHT layer. Figure 6 compares the relationship between $O_3$ and $NO_x$ at PD and SHT. The results, considering only nonrainy periods, are displayed in Figure S2, exhibiting little difference from Figure 6 due to the low solubility of $O_3$ and $NO_x$. As VOC measurements are not available in this study, the observed $O_3$-$NO_x$ relationship can reflect the relative role of these two major precursors in the formation of $O_3$. The observational evidence shows that $O_3$ and $NO_x$ concentrations present negative correlations in cold months at both PD and SHT, suggesting that $O_3$ production is inhibited by increased $NO_x$ concentrations. The negative $O_3$-$NO_x$ correlation elucidates why the wintertime $O_3$ production is not significant at SHT, even during the daytime when the surface $NO_x$ can be transported upward by the weak vertical mixings. In addition to high $NO_x$ emissions [60], the atmospheric composition at SHT is more affected by air masses from north China in cold months, according to the back trajectory results (Figure 7). The influx of polluted north winds carries $NO_x$-saturated air, which increases the ambient $NO_x$/VOCs ration and thereby reinforces the $NO_x$-saturated $O_3$ formation [7,61]. In warm months, the negative $O_3$-$NO_x$ relationship is observed to be weak. Instead, $O_3$ concentrations tend to exhibit positive correlations with $NO_x$, with the most significant positive correlations observed in July. Compared to those near the surface, $O_3$ concentrations at SHT present a more sensitive response (2.45 ppb$^{-1}$) to $NO_x$ changes, indicating that $O_3$ formation is more limited by $NO_x$ in the upper layer. The result explains why a similar magnitude of $NO_x$ results in more $O_3$ production at SHT than PD, and the observed large OD during the summertime. The back trajectory results suggest that the influence from southwest winds intensifies during the summertime, accounting for 50–55% of the total flows affecting the SHT region. These southwest flows have been proved to be abundant in biogenic VOCs as they traverse extensive forested areas, which are conducive to the production of hydrogen radicals in warm months [62].

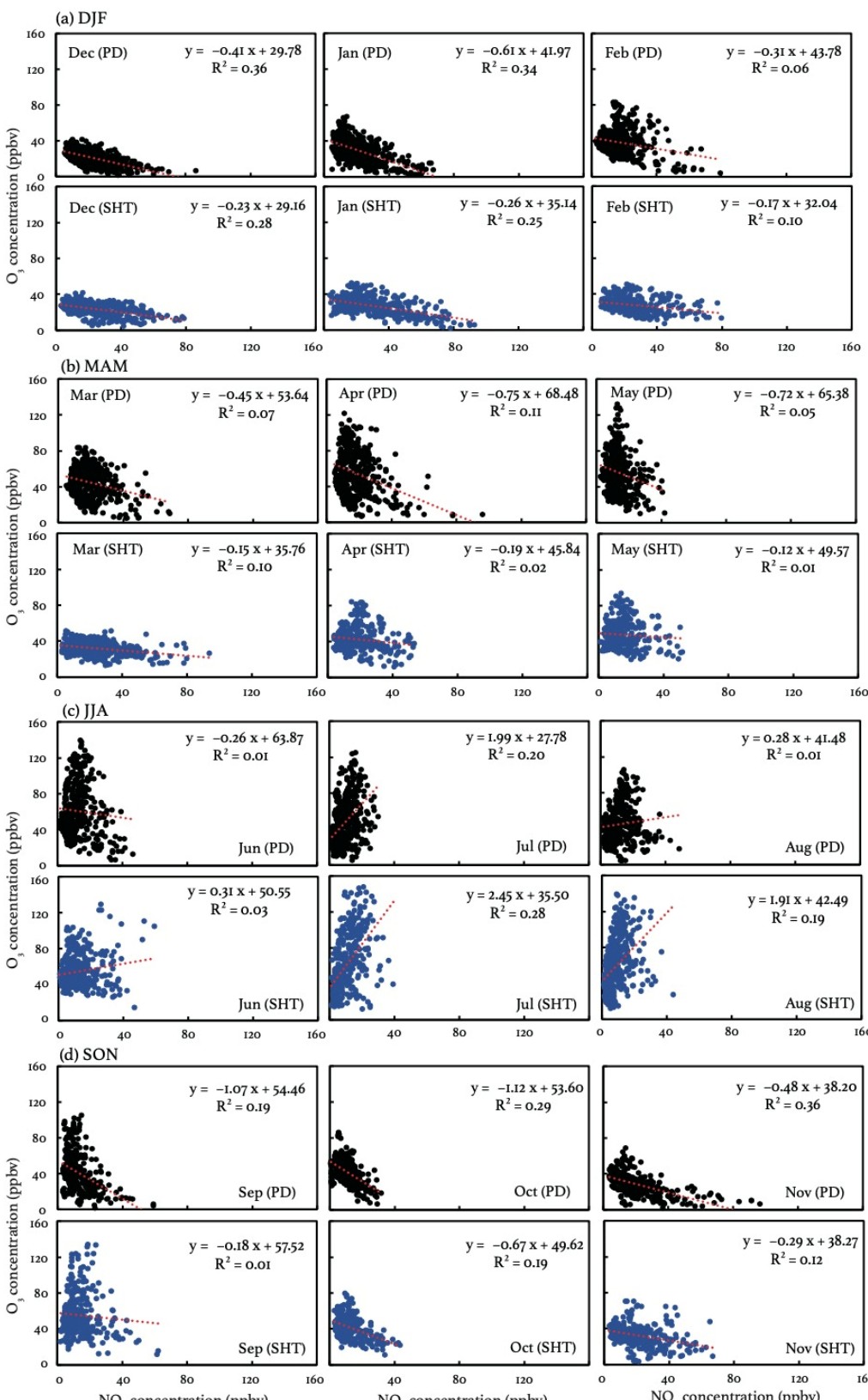

**Figure 6.** The scatter plots of monthly mean $O_3$ (*y* axis) and $NO_x$ (*x* axis) concentrations during the daytime (09:00–16:00 LST) in (**a**) DJF, (**b**) MAM, (**c**) JJA, and (**d**) SON during 2017–2018.

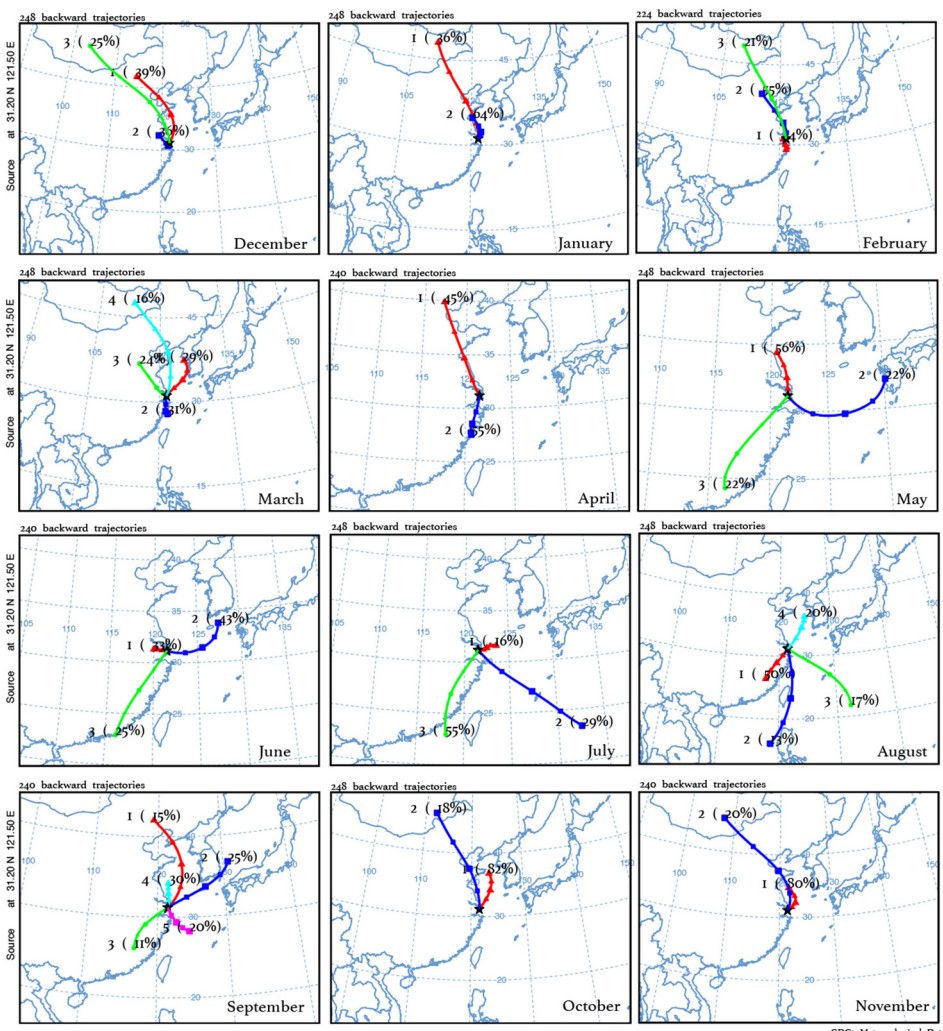

**Figure 7.** The 48 h backward trajectory cluster results during each month in 2017 and 2018 using the Hybrid Single-Particle Lagrangian Integrated Trajectory (HYSPLIT) model driven by the reanalysis data ($2° \times 2.5°$) provided by the National Centers for Environmental Prediction (NCEP) and the National Center for Atmospheric Research (NCAR). The starting location is where SHT locates (31.2° N, 121.5° E, 600 m) the percentage of each group in the total number of flows are marked in each figure.

## 4. Conclusions

In this study, we characterize $O_3$ variations at a high-altitude monitoring site, SHT, based on measurements from a 2-year continuous period and discuss the potential impacts of the vertical exchange of air pollutants on the $O_3$ air quality within the urban PBL in Shanghai. SHT is located in the ML during the daytime and above the NBL during the nighttime, where the observations provide a unique proxy to study the different characteristics of $O_3$ and its precursors in different PBL structures.

The daytime $O_3$ concentrations at SHT are observed to be higher than the surface ones in warm months, indicating that the vertical exchange tends to result in downward $O_3$ flux, thereby aggravating surface pollution. During this period, vertical mixing is observed to be more efficient, resulting in a substantial amount of surface $O_3$ precursors transported upward to the SHT layer. The $O_3$ formation at SHT is observed to more limited by $NO_x$ compared to the surface, resulting in increased $O_3$ production when a similar amount of $NO_x$ is presented. The prevailing southwest winds which are abundant in biogenic VOCs contribute to the $NO_x$-limited formation in the urban PBL as well. In contrast, the daytime $O_3$ at SHT is observed to exhibit lower levels than the surface $O_3$ in cold months. During

this period, the $O_3$ formation is observed to be inhibited by $NO_x$ increases at both PD and SHT. As the vertical mixings are weak, $O_3$ at SHT thus exhibits little response to surface sources. The daytime vertical mixings are more likely to mitigate surface $O_3$ pollution due to the negative vertical gradient of $O_3$.

Additionally, the observations provide direct evidence of the $O_3$ entrainments in the morning. A nighttime $O_3$ reservoir layer and consistent negative OCR during 05:00–07:00 LST at SHT are detected all year round. Limited by the means of these measurements, we might not be able to quantify the impacts of these vertical exchanges on surface air quality based on the current dataset. However, our results provide observational evidence that vertical mixings, either in the morning or during the daytime, can occur within the initial layer (~600 m) of air quality models. To enhance our understanding of the vertical transport processes of $O_3$ and other related species, additional measurements in different boundary layers, encompassing a broader range of species and meteorological parameters, are essential. There is also an increasing need for improved vertical resolution under 900 mb in air quality models to better represent these processes.

**Supplementary Materials:** The following supporting information can be downloaded at: https: //www.mdpi.com/article/10.3390/atmos15030248/s1, Figure S1: Similar results as Figure 3 but with measurements only during nonrainy periods; Figure S2: Similar results as Figure 6 but with measurements only during nonrainy periods.

**Author Contributions:** Conceptualization, Y.G. and J.X.; methodology, Y.G. and J.X.; validation, F.Y., J.X., C.Y., L.P. and W.G.; formal analysis, Y.G. and J.X.; investigation, Y.G. and J.X.; resources, J.X.; data curation, F.Y., C.Y., L.P. and W.G.; writing—original draft preparation, Y.G.; writing—review and editing, Y.G., J.X. and H.L.; visualization, Y.G.; project administration, J.X.; funding acquisition, Y.G. and C.Y. All authors have read and agreed to the published version of the manuscript.

**Funding:** This research was funded by the National Natural Science Foundation of China, grant number 42205189, Natural Science Foundation of Shanghai, grant number 22ZR1467500, and Shanghai Sailing Program, grant no. 21YF1412400.

**Institutional Review Board Statement:** Not applicable.

**Informed Consent Statement:** Not applicable.

**Data Availability Statement:** The data presented in this study are available on request from the corresponding author. The data are not publicly available due to the data protection regulation of Shanghai Meteorological Service.

**Acknowledgments:** We acknowledge all the anonymous reviewers for their comments to improve the original manuscript.

**Conflicts of Interest:** The authors declare no conflicts of interest.

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
