# Peer review of "Observational Evidence of the Vertical Exchange of Ozone within the Urban Planetary Boundary Layer in Shanghai, China"

_atmosphere, doi:10.3390/atmos15030248_

Round 1
Reviewer 1 Report
Comments and Suggestions for Authors
The observations at SHT and ground stations provide unique data for this study. However, there are a lot of confusions in this study. First, there are no labels and physical units for most of the figures. Second, How are the data selected for the seasonal average? Because the O3-NOx chemistry mainly happens in the daytime under sunny day, I would suggest on selecting the O3-NOx data in the daytime (8:00-18:00 Beijing time) and sunny skies. The data in the night and raining day seem noisy data for the O3-NOx chemical analysis. In addition, please cite the references that studies PBL diurnal variation from the lidar and radiosonde observations, these may help judge whether the SHT height is within PBL or not. Please see the comments below in details.
1. Line 140-159, There are several questions on the modeling PBL-height that is relevant whether the SHT height is located in the PBL or not. At noon time, the SHT is reasonably located in the PBL. However, in the early morning and night, the SHT may be above the PBL or within the PBL depending on the PBL height (PBLH).
Please cite the references that directly discuss or analyze the PBL height observations in Shanghai such as the lidar observations in Shanghai, radiosonde observation around Shanghai, please do not show the references at other region. This help judge the PBLH dynamics in Shanghai.
2. Line 162-169, these discussions should be moved to the “Methodology”. They are neither your data nor results.
3. Fig.2, both X-axis and Y-axis are not labeled. What are these number and their physical units?
The PBLHs are highest in October (Fig.2c) than summer month (Fig.2b). It seems not reasonable. Can you plot the radiosonde data in Shanghai area or compare to other observations of PBLH in Shanghai? The solar radiances are generally lower in October than the summer months (June, July and August).
4. Fig.3, both X-axis and Y-axis are not labeled. What are these number and their units? Figure 3, are these value averaged with the data under rains and clouds? Since the O3 production is sensitive to the solar UV radiance and temperature, the rains and clouds can definitely affect the O3 concentration at SHT and PD.
5. Line 34-341, lack of O3 precursor emission sources at SHT (600-m)? The NOX and VOCs can be transported to 600-m altitude, right? Some high rising buildings can emit the NOx and VOCs, right?
6. Fig.6, please explain why the NO2 at SHT in DJF and MAM seasons (Fig.6a and Fig.6d) are much higher at 10:00-16:00 than those at PD site?
Are these seasonal average including all the data under raining days and other days?
7. Fig.7. What time period of the data are used for these correlation analysis? In order to observe the chemical relationship of NOx-O3, please focus on the correlation analysis in the daytime (10:00-18:00 Beijing time) under the clear sky data. Please don’t show the data in the night and under the mostly cloudy and raining days.
8. Line 387-388 and Fig.8, the HYSPLIT model trajectories only indicate the airmass transport path. In fact, all the pollutants including NOx, VOCs and O3 can be transported together. You can not only attribute to NOx transport issue.
Do you think that the emissions of NOx and VOCs from Shanghai are less than its surrounding area when the study argued the pollution transport from the surrouding area of Shanghai?
Need revisions on the writing style.
Reviewer 2 Report
Comments and Suggestions for Authors
In the article the seasonal variations of average monthly ozone concentrations are analyzed at two levels of the atmospheric boundary layer - at the near ground surface and at the height of about 600 meters (Shanghai Tower). Direct year-round measurements of ozone, as well as its probable precursors, at such altitudes are extremely rare, and this is the main novelty and value of the presented research. This allowed the authors to assess the influence of seasonal processes in the atmospheric boundary layer (variability of its height, vertical mixing) on air pollution in the surface layer. Unfortunately, the authors limited themselves to analyzing only the average monthly concentrations of ozone and nitrogen oxides; it would be extremely interesting to see episodes of higher time resolution (minutes to hours) of such fluctuations. Most likely, this could better represent the processes of vertical mixing of impurities and chemical interaction between impurities. There is also a lack of data on meteorological parameters near the ground and at altitudes. Perhaps the authors will be able to cover the topic of shorter-period variations in a separate article. Despite this shortcoming, the article as a whole is worthy of publication, taking into account some of the comments that are presented below.
Some recommendations for the text.
Line 16-17: in the text - “O3 variations at a high-latitude monitoring site…” the authors probably meant the “high location” or “high altitude”, but not latitude, because the latitude of Shanghai is not high – it is closer to Equator than to Pole. Same correction is needed for line 414 (in Conlusion).
Line 33: “Surface O3 is generated from nitrogen oxides (NOx=NO+NO2)” – this formula does not show how ozone is formed. Moreover, in the following sentences (lines 36-37 as well as during all text) this statement is refuted by the authors themselves - “However, the sharply reduced emissions of NOx led to an unexpected increase in surface O3 levels [6–8]”.
It would be desirable to describe this contradiction more clearly (theoretically in natural reactions, nitrogen compounds can participate in the generation of ozone, but anthropogenic nitrogen oxides usually suppress ozone).
Figure 4. On the reviewer’s opinion, the right side of the Figure 4 ("O3 difference (OD)") is not of interest - it is the same data as at the left side. It would be more interesting to put here the NOx graphs from Figure 6 (to combine Fig 4 and 6).
Reviewer 3 Report
Comments and Suggestions for Authors
General comment
The paper aims to understand the O3 vertical transport caused by the variation in the PBL height. The author identified that the residual layer could hold the O3 and its precursor, causing peaks at the ground level.
The paper is interesting because vertical transport is crucial for understanding the O3 concentration across urban centers, and the PBL plays a vital role in this concentration. The paper suits other researchers interested in O3 formation, modeling, and measurements.
After addressing the small comments in the Materials and Methods and Results and Discussions, I recommend this paper for publication.
Introduction
The introduction explained the physics of the PBL evolution and how it impacts the O3 concentration near the surface. Additionally, the author mentions the lack of measurement in the upper PBL, especially at night, exposing the gap they want to fill.
Materials and Methods
The methodology is concise and straight to the point.
Why did the author use the average height of the two years of the PBL and not the daily information? The ERA5 has the daily PBL height. Please reinforce the explanation and justify this in the text.
Results and Discussions
The results are well-written and well-explained.
Figure 3 should have the correct hours in the tick labels. It is better for the reader and for presentation. Please correct it.
Conclusions
The conclusion is also well-written and summarizes the findings of the paper.
Round 2
Reviewer 1 Report
Comments and Suggestions for Authors
Thanks for the revisions